# Associations between Preoperative Patient Socioeconomic Status and Pain-Related Outcomes with Pain and Function in Patients Undergoing Rotator Cuff Repairs

**DOI:** 10.3390/healthcare11202786

**Published:** 2023-10-21

**Authors:** Borja Perez-Dominguez, Sara Perpiña-Martinez, Sara Garcia-Isidoro, Isabel Escobio-Prieto, Alvaro Manuel Rodriguez-Rodriguez, Maria Blanco-Diaz

**Affiliations:** 1Exercise Intervention for Health Research Group (EXINH-RG), Department of Physiotherapy, University of Valencia, 46010 Valencia, Spain; f.borja.perez@uv.es; 2Faculty of Nursing and Physiotherapy Salus Infirmorum, Pontifical University of Salamanca, 28015 Madrid, Spain; 3Fisioterapia García Isidoro, 28024 Madrid, Spain; saragey@gmail.com; 4Institute of Biomedicine of Seville (IBIS), Department of Physiotherapy, Faculty of Nursing, Physiotherapy and Podiatry, University of Seville, 41004 Seville, Spain; iescobio@us.es; 5Faculty of Medicine and Health Sciences, Physiotherapy and Translational Research Group (FINTRA), Institute of Health Research of the Principality of Asturias, University of Oviedo, 33003 Oviedo, Spain; alvaro.manuel.rodriguez@gmail.com (A.M.R.-R.); blancomaria@uniovi.es (M.B.-D.)

**Keywords:** rotator cuff injuries, pain measurement, socioeconomic factors, health status disparities, shoulder pain, functional outcome assessment, cross-sectional studies

## Abstract

BACKGROUND: Patients undergoing rotator cuff repairs commonly experience postoperative pain and functional limitations. Various socioeconomic and pain-related factors have been recognized as influential in the prognosis of such patients. This study aims to investigate the associations between postoperative pain and functionality and preoperative pain-related outcomes and socioeconomic status in patients undergoing rotator cuff repairs. METHODS: This cross-sectional study examines the relationship between the outcomes of rotator cuff repairs and participants’ socioeconomic status and pain-related measures. Socioeconomic status was assessed through indicators such as educational level, monthly household income, and occupation. Pain-related outcomes included measures of kinesiophobia and pain self-efficacy. RESULTS: A total of 105 patients (68 male, 37 female) were included in the analysis. The findings revealed no significant association between postoperative pain or functionality and the patients’ socioeconomic status (*p* > 0.05). However, postoperative pain levels demonstrated a significant association with preoperative kinesiophobia (*p* < 0.05) and pain self-efficacy (*p* < 0.013). In contrast, functionality did not exhibit a significant association with these measures (*p* < 0.072 and 0.217, respectively). CONCLUSIONS: Preoperative pain-related outcomes play a role in postoperative pain levels among patients undergoing rotator cuff repairs. However, they do not appear to be related to functionality. Additionally, socioeconomic status does not significantly impact either pain or functionality.

## 1. Introduction

Rotator cuff injuries account for approximately 80% of primary care consultations for shoulder pain [1], and often result in associated weakness and loss of function. Despite the development of effective management strategies, around 30% of patients with rotator cuff injuries report no improvement in pain or functionality [2,3]. This highlights the importance of addressing not only the underlying factors causing pain but also the associated factors that can significantly influence a patient’s perception of pain.

The emerging biopsychosocial approach to managing patients in painful clinical settings emphasizes the consideration of both injury-related factors and confounding factors that may influence the patient’s pain perception and coping abilities [3,4]. Several preoperative factors have been identified as potentially affecting the outcomes of rotator cuff repairs. These factors include patient clinical characteristics [4], emotions and cognitions related to pain [5], and psychological status [6]. Clinical characteristics associated with prognosis include receiving workers’ compensation [7], muscle strength [2], sex, and race [4]. Among the emotional and cognitive predictors, those related to the pain experience have been linked to persistent postoperative shoulder pain [8]. Kinesiophobia, characterized by a fear of pain caused by movement, has been associated with higher levels of pain and functional impairment in shoulder disorders, although the results on this relationship are conflicting [9]. Furthermore, factors related to a patient’s ability to perform activities, such as pain self-efficacy, have been associated with better prognoses and improvements in pain and function in patients with persistent musculoskeletal pain [10,11]. Previous studies have described various preoperative factors that influence patient outcomes after shoulder surgery. Negative cognitive–affective responses to pain have been shown to impact functional outcomes in patients undergoing shoulder arthroscopy [12], highlighting its significant effect on treatment success. Psychological factors, including depression and anxiety, have also been correlated with poorer pain and functionality outcomes [6].

A wide range of instruments is available to assess preoperative factors. Pain intensity assessments commonly use the Visual Analogue Scale (VAS) [13,14,15,16], due to its simplicity and ease of understanding. For evaluating shoulder function, assessment tools range from more general methods, such as the American Shoulder and Elbow Surgeons Standardized Shoulder Assessment Form [17], to more specific measures tailored to particular samples or conditions, such as the Western Ontario Rotator Cuff (WORC) index [1] for rotator-cuff-related injuries.

Various instruments are used to assess preoperative factors that influence pain perception and the pain experience. For example, the development of fear of movement, or kinesiophobia, is considered a significant predictor of pain behavior, and the Tampa Scale of Kinesiophobia (TSK) [18] is widely used to assess this construct. Another important construct involves a patient’s ability to perform activities despite pain, commonly known as self-efficacy. Several tools have been developed to assess self-efficacy, including condition-specific scales such as the Arthritis Self-Efficacy Scale (ASES) and the Chronic Disease Self-Efficacy Scale, as well as those specifically related to pain and altered pain states, such as the Chronic Pain Self-Efficacy Scale or the Pain Self-Efficacy Scale (PSEQ) [19,20]. The PSEQ is the preferred scale for clinicians when dealing with clinical settings involving musculoskeletal disorders. It consists of a 10-item questionnaire originally developed in English to assess the confidence and ability of individuals with pain to engage in activities despite their pain [21].

The socioeconomic status of an individual encompasses various factors that influence multiple domains of their life. Previous studies have established a correlation between a patient’s socioeconomic status and their perception of pain. It has been shown that patients who reside in more affluent areas [22], who have lower educational levels [23], or who experience low income levels or financial strains [24] tend to exhibit higher-prevalence levels of pain [25]. Socioeconomic status is typically assessed using the Socioeconomic Status Questionnaire [26,27], an adaptive instrument that employs simple questions to determine different levels within various categories. When evaluating the socioeconomic status of a patient, common domains considered include educational level, household monthly income, and job rank. These factors have previously been taken into account when assessing the outcomes of interventions for rotator cuff surgical repairs [4].

Identifying the factors that directly influence the prognosis of patients with rotator cuff injuries can facilitate outcome prediction and aid in decision making. Therefore, the primary objective of this study is to analyze changes in pain intensity and functionality among patients undergoing rotator cuff surgical repairs and assess the correlation between preoperative pain-related outcomes and the socioeconomic status of the patients.

## 2. Materials and Methods

### 2.1. Design

This is a cross-sectional study that examines the associations between preoperative and postoperative surgical outcomes. Participants were informed about the study’s objectives and provided written consent prior to enrollment. The study adhered to the ethical principles outlined in the Declaration of Helsinki and received approval from the Ethics Committee of the University of Valencia (registry number: 2537378, approval date: 9 March 2023).

### 2.2. Participants

Participants were recruited from a private practice hospital in Valencia, Spain, between the months of March and May 2023. The assessments took place before the surgery and three weeks after the surgery, through the use of self-administered questionnaires. A member of the research team assisted participants in filling out the questionnaires. Potential participants attended a medical consultation where they were provided with a detailed explanation of the study’s procedure by a research team member. Following that, surgeons evaluated each participant’s clinical presentation and made a professional medical decision on whether surgical repair was necessary. Participants who were scheduled for surgery and gave written consent were enrolled in the study. On the other hand, participants referred for conservative management were directed to Physical Therapy and were considered “on hold” in case conservative management failed and surgery became necessary. In such cases, they would then be eligible for the study. A member of the research team collected all the responses and recorded the results in a spreadsheet for further analysis.

### 2.3. Outcomes

Each outcome was self-administered by the participant, with the assistance of a member from the research team if the participant encountered any difficulty in understanding specific items in the questionnaires. Participant characteristics, including age, sex, smoking status (non-smoker, former, or current), and whether the injured shoulder was the dominant arm or not, were recorded preoperatively by a member of the research team.

#### 2.3.1. Pre- and Post-Surgical Outcomes

Pre- and post-surgical assessments were conducted by a member of the research team to evaluate the participant’s pain intensity using the VAS and their self-reported functionality using the WORC. The VAS is a 100 mm horizontal scale that ranges from no pain at all to the maximum possible pain [28]. Participants were asked to mark their perceived pain level on the horizontal line at the time of assessment. The WORC is a reliable and valid disease-specific tool designed for patients with rotator cuff injuries. It consists of 21 items that assess five different domains: physical symptoms (including pain), sports and recreation, work, lifestyle, and emotions. Each item is scored using a 100 mm horizontal scale, with higher scores indicating worse outcomes. A Spanish translation of the original validated WORC questionnaire was used [1].

#### 2.3.2. Outcomes for the Correlation Analysis

Assessments for the correlation analysis included participants’ fear of movement, pain self-efficacy, and outcomes regarding their socioeconomic status. Fear of movement was evaluated using the TSK, a scale that assigns scores ranging from 17 to 68, with higher scores indicating a greater fear of pain, movement, and injury. A translation into Spanish of the original TSK was used in this study. Participants’ pain self-efficacy was assessed using the PSEQ, a 10-item questionnaire originally developed in English by Albert Bandura in 1978 [29] for individuals with chronic low back pain. The PSEQ aims to gauge the confidence and ability of individuals with pain to engage in activities despite their pain. This questionnaire considers the pain-processing experience and the relationship between pain and self-efficacy. Scores on the PSEQ range from 0 to 60, with higher values indicating stronger self-efficacy levels [21]. Previous studies have utilized the PSEQ to assess patients with various chronic pain conditions, and its validity and utility in evaluating pain related to rotator cuff injuries have also been investigated [30,31]. The Spanish validated version of the PSEQ was utilized for this study [32].

The socioeconomic status of the participants was assessed using the Socioeconomic Status questionnaire [5,27]. This adaptive instrument categorizes participants into different levels within a predetermined range by asking simple questions about various domains related to their socioeconomic status. For this study, participants were asked about their educational level, household monthly income, and job rank, which are commonly used domains when assessing socioeconomic status. Educational level was determined using the International Standard Classification of Education (ISCED), which consists of nine different levels ranging from 0 (pre-scholar educational level) to 8 (doctorate educational level). Household monthly income levels were established using the annual survey conducted by the Spanish National Institute of Statistics (INE), with 11 different levels ranging from 0 to more than EUR 5000 per month. Job rank was determined based on the latest collective occupational agreement from the Spanish Ministry of Labor, which includes six different levels. Level 1 represents a job that does not require a degree, while level 6 corresponds to a job that necessitates a college-level degree.

### 2.4. Statistical Analysis

Statistical calculations were conducted using SPSS Software v.23 for MacOS (IBM, Chicago, IL, USA). The sample size determination was performed using G-Power software v.3.1.9.3 (G-Power, Aichach, Germany), taking into consideration previous studies [33,34] with similar objectives and assessing the clinical significance of their observed effects. A minimum effect size of 0.25 was deemed appropriate for this study. The level of statistical power was set at 80% with an α-error of 0.05, resulting in a required sample size of at least 100 participants to achieve the desired level of statistical power. The Kolmogorov–Smirnov test was employed to assess the normal distribution of the data. Results were presented as mean and standard deviation or as count and percentage, depending on the nature of the outcome. Imputation was anticipated as the approach for handling missing data.

A one-way ANOVA was performed to analyze the change over time in VAS and WORC scores, and the results were reported as mean differences and standard deviations for each outcome. These results were utilized to conduct correlation analyses involving participant characteristics, TSK, PSEQ, and socioeconomic outcomes. Pearson’s correlation coefficient was used to establish the correlation analysis between the results and TSK and PSEQ outcomes. For the correlation analysis between the results and socioeconomic outcomes, the non-parametric test Spearman’s rho was employed. The correlation coefficient and *p*-value were reported for each correlation analysis. The correlation coefficient values range from −1 to 1, indicating the strongest possible negative and positive correlations, respectively, while values close to 0 indicate weak correlation levels. Significance was determined at a *p*-value threshold of 0.05.

## 3. Results

### 3.1. Participants

A total of 105 participants were assessed. Every participant completed their assessments prior to the surgery and 3 weeks after the surgery. No missing data were found. Their baseline demographic characteristics, smoking status, injured shoulder dominance, values for TSK and PSEQ, and values for their socioeconomic status are presented in Table 1. Participants’ mean age in this study was 57.4 (SD 10.2) years; they were predominantly male (68 participants, 65%) and non-smokers (83 participants, 79%) and sustained their injured rotator cuff in their dominant upper limb (74 participants, 70%). Overall values showed the presence of kinesiophobia (mean 47.0, SD 9.5), and participants presented moderate levels of pain self-efficacy (mean 42.5, SD 11.2). Every participant had an educational status of at least level 4, with most of them concentrated at level 7 (37 participants, 35%), which is equivalent to college-level education; household monthly income levels were in the medium-range levels between EUR 1000 and 3400, with most participants reporting income levels of EUR 1500 to 1999 (23 participants, 22%); and, despite many participants having a job that required no degree at all (22 participants, 21%), most of them had jobs that required a college degree, at levels 5 and 6 (72 participants, 68%).

### 3.2. Results from the ANOVA

The results of the ANOVA analysis for VAS and WORC values showed significant improvements (*p* < 0.001) in the overall sample and the stratified groups by sex. The VAS scores improved, in the overall sample, with a mean difference of 3.7 points (SD 1.5 points), from 8.3 (SD 1.7) points preoperatively to 4.6 (SD 1.8) points postsurgery; in men, with a mean difference of 3.4 points (SD 1.6 points), from 7.8 points (SD 1.5 points) to 4.4 points (1.7 points); and, in women, with a mean difference of 4.1 points (SD 1.2 points), from 8.8 points (SD 1.7 points) to 4.7 points (SD 1.7 points). The WORC scores, in the overall sample, improved with a mean difference of 30.2 points (SD 17.6 points), from 63.4 (SD 19.1) points preoperatively to 33.2 (SD 17.2) points postsurgery; in men, with a mean difference of 31.8 points (SD 14.2 points), from 62.2 points (SD 15.0 points) to 30.4 points (SD 14.5 points); and, in women, with a mean difference of 28.2 points (SD 16.6 points), from 64.8 points (SD 13.8 points) to 36.6 points (SD 17.2 points). The results are shown in Table 2.

### 3.3. Correlation Analysis

The results from the correlation analysis conducted between the VAS and WORC results after the surgery with the preoperative TSK and PSEQ scores, and the participant’s educational level, household monthly income, and job rank, for the overall sample and the stratified groups by sex, are shown in Table 3.

#### 3.3.1. Correlation between Results and Fear of Movement and Pain Self-Efficacy

The correlation analysis between pain intensity and fear of movement yielded significant moderate-to-strong positive levels of association for the overall sample, men, and women (*p* < 0.05), meaning that higher values for pain intensity were correlated with higher values of fear of movement or kinesiophobia. Pain intensity and pain self-efficacy showed significant negative moderate-to-strong correlation levels for the overall sample and for women, meaning that higher pain intensity values were correlated with lower scores in the PSEQ, which meant worse outcome measures in pain self-efficacy. In men, there were low levels of association and they were non-significant.

The correlation analysis between function and fear of movement showed non-significant positive low levels of association in the overall sample, men, and women, meaning that higher scores for functionality, that indicate worse outcomes, were correlated with higher scores for fear of movement or kinesiophobia. Function and pain self-efficacy showed non-significant negative low levels of association for the overall sample and men; however, significant negative moderate-to-strong levels of association were found in women, meaning that higher outcome scores in functionality, which indicate worse outcomes, showed a correlation with lower scores in the PSEQ, indicating worse self-efficacy.

#### 3.3.2. Correlation between Results and Socioeconomic Status Outcomes

The correlation analysis between the results and participants’ socioeconomic status showed that there were non-significant association levels between pain intensity and function, and participants’ educational level, household monthly income, or job rank in the overall sample, men, or women. Even though non-significant correlations were found, the association levels between pain intensity and the participant’s educational level showed a low negative correlation in the overall sample, men, and women, meaning that higher pain intensity outcomes were correlated with lower educational levels. The same low negative association was found between pain intensity and household monthly income for the overall sample, men, and women, meaning that higher levels of pain were correlated with lower levels of household monthly income. The association levels between pain intensity and job rank showed a low positive correlation for the overall sample, men, and women, meaning that higher levels of pain intensity were correlated with higher levels of job rank. Finally, low positive correlation levels were also found between functionality and every socioeconomic status domain for the overall sample, men, and women, meaning that higher scores in functionality, which indicate worse outcomes, were correlated with higher educational levels, household monthly income, and job rank levels.

## 4. Discussion

The primary objective of this study was to examine the changes in pain intensity and functionality among patients who underwent surgical repairs for rotator cuff injuries. Additionally, we aimed to assess the correlation between these changes with preoperative pain-related outcomes and the socioeconomic status of the patients. The results revealed significant associations between postoperative pain intensity and preoperative pain-related outcomes, specifically in relation to kinesiophobia assessed using the TSK scale and pain self-efficacy measured with the PSEQ questionnaire. However, no significant associations were observed between postoperative functionality and preoperative outcomes. Furthermore, there were no associations found between postoperative pain and functionality, and the participant’s educational level, household monthly income, or job rank. It is important to note that, while the surgical management of rotator cuff injuries leads to substantial improvements in pain and functionality, the prognosis can be influenced by various factors, including psychological [5,6], physical [2], and social [4,7] factors.

The analysis revealed significant moderate-to-strong correlations between the VAS results and preoperative pain-related outcomes. On the other hand, the correlations between the WORC results and preoperative pain-related outcomes were found to be non-significant and ranged from low to moderate. These findings align with previous studies, including the work conducted by Suer et al. [8], which showed a positive association between TSK scores and postoperative pain levels even three months after surgery. These results suggest that the association between kinesiophobia, as measured via the TSK, and prolonged pain following shoulder surgery may persist over time, indicating the potential value of kinesiophobia as a valid predictor for long-term pain outcomes.

In our study, no significant association was found between patients’ educational levels and postoperative outcomes. This differs from the findings of Harris et al. [3], who reported a positive association between education levels and WORC values, and Dunn et al. [35], who found a positive association with pain levels. However, the disparity in results could be attributed to the differences in sample characteristics. Harris et al. and Dunn et al. alike had more diverse educational levels in their samples, whereas our study had a predominant concentration of participants with higher educational levels. It is plausible that lower educational levels might have influenced the potential correlation between these outcomes. Additionally, Dorner et al. [25] showed an inverse correlation between educational level, income, job rank, and the prevalence and intensity of pain, and subjective feeling of disability. However, it is important to note that their sample was not specific to rotator cuff injuries, and pain location could be a contributing factor in establishing association levels.

Furthermore, there was no association observed between postoperative pain and functionality outcomes and the participants’ household monthly income. The majority of participants in our study fell within the medium-range income levels, indicating that they were not facing significant financial difficulties. This lack of correlation with postsurgical prognosis can be explained by their relatively stable financial situation. In contrast, several studies conducted by Aggarwal et al. [22], Latza et al. [23], or Hagen et al. [24] have reported a relationship between financial struggles and prognostic values. However, it is important to note that these studies were conducted in different countries (Norway, Germany, Sweden), where the financial landscape may differ substantially from that of our study conducted in Spain. Additionally, these studies specifically included participants classified as experiencing “financial struggle”, which was not the focus of our study. Future studies should consider incorporating participants with low household monthly income levels to determine the association between this domain and postsurgical pain and functionality.

Moreover, there was no association found between job rank and postoperative outcomes for pain and functionality. Despite the diversity in job ranks among participants, with 21% having jobs that required no degree and 68% having jobs that required a college-level degree, no significant correlation was observed. This suggests that job category alone may not be a relevant factor, and the correlation might instead be related to occupational status or the extent to which the participant’s job involves significant use of the shoulder. In a study by Sahoo et al. [4], occupational status was found to be correlated with patient-reported outcome measures in 1442 patients who underwent rotator cuff repair surgery, indicating that being employed or unemployed is relevant in determining postoperative prognosis. Furthermore, regardless of the degree required for a particular job, the degree of shoulder involvement in job performance could be a relevant consideration. For example, a surgeon or a baker may exert considerably more strain on the shoulder compared to a receptionist or a college professor. These factors should be considered in future correlation analyses.

This study has several limitations that should be acknowledged. Firstly, there is a potential for selection bias despite the inclusion of a considerably large sample. Participant recruitment and enrollment were conducted exclusively at a private practice hospital in Valencia, Spain, which limits access to only those patients with private insurance. This could introduce a bias in the interpretation of association levels with socioeconomic status, as the study included participants predominantly from a relatively high socioeconomic background. Furthermore, the choice of outcomes for the correlation analysis could have been expanded to include participant demographic characteristics, such as sex, which have been shown to correlate with patient prognostic values [4]. Additionally, the assessments were conducted only 3 weeks after the intervention, and it remains to be determined whether longer follow-up assessments would yield similar association levels. Also, this study assessed fear of movement through a translated version in Spanish of the original TSK rather than the Spanish validated version [35], which includes 11 items instead of the original 17, so the results must be cautiously interpreted, as different versions of the same scale might yield different results. A Spanish translation of the original WORC index [1] was used, too, as the authors were only able to find validated Spanish versions in other cultural settings, such as Colombia [36], where trans-cultural influences might have been relevant. Future studies should investigate the influence of socioeconomic status on postsurgical outcomes in different populations with a more heterogeneous distribution.

Nevertheless, our study boasts several notable strengths. To the best of our knowledge, it is the first investigation that examines pain self-efficacy as a potential predictive factor associated with postoperative outcomes in rotator cuff repairs. Additionally, it is the first study to explore the significance of household monthly income and job rank as relevant socioeconomic factors that may exert an influence. These findings lay the foundation for future assessments, empowering healthcare professionals to effectively identify prognostic indicators for outcomes in rotator cuff repairs, thereby enhancing the management of these conditions. Furthermore, despite the self-administration of questionnaires, the inclusion of patient assessments during medical consultations yielded exceptional response rates and eliminated missing data concerns.

## 5. Conclusions

The results of this study indicate that kinesiophobia and pain self-efficacy have a significant impact on postoperative pain levels in patients undergoing rotator cuff repairs. However, these factors do not exhibit a significant influence on functionality. Moreover, participants’ educational levels, household monthly income, and job ranks do not show any significant associations with postoperative pain or functionality. Therefore, it can be concluded that, while kinesiophobia and pain self-efficacy play a role in postoperative pain, other socioeconomic factors do not appear to be significant predictors in this context.

## Figures and Tables

**Table 1 healthcare-11-02786-t001:** Participants’ baseline characteristics, preoperative outcomes, and socioeconomic status.

Outcome	Mean (SD) or Count (%)
Age (years)	57.4 (10.2)
Sex	
Male	68 (65%)
Female	37 (35%)
Smoking status	
Non-smoker	83 (79%)
Former	5 (5%)
Current	17 (16%)
Injured shoulder	
Dominant	74 (70%)
Non-dominant	31 (30%)
Tampa Scale of Kinesiophobia	47.0 (9.5)
Pain Self-Efficacy Questionnaire	42.5 (11.2)
Educational level	
Level 0	0 (0%)
Level 1	0 (0%)
Level 2	0 (0%)
Level 3	0 (0%)
Level 4	13 (12%)
Level 5	21 (20%)
Level 6	29 (28%)
Level 7	37 (35%)
Level 8	5 (5%)
Household monthly income (EUR)	
0–499	3 (3%)
500–999	0 (0%)
1000–1499	20 (19%)
1500–1999	23 (22%)
2000–2499	17 (16%)
2500–2999	15 (14%)
3000–3499	22 (21%)
3500–3999	2 (2%)
4000–4499	1 (1%)
4.500–4.999	0 (0%)
>5.000	2 (2%)
Job rank	
Level 1	22 (21%)
Level 2	3 (3%)
Level 3	2 (2%)
Level 4	6 (6%)
Level 5	40 (38%)
Level 6	32 (30%)

**Table 2 healthcare-11-02786-t002:** Results of the ANOVA.

Outcome		Mean PRE (SD)	Mean POST (SD)	Mean Difference (SD)	*p*-Value
Overall	VAS	8.3 (1.7)	4.6 (1.8)	3.7 (1.5)	<0.001 *
	WORC	63.4 (19.1)	33.2 (17.2)	30.2 (17.6)	<0.001 *
Men	VAS	7.8 (1.5)	4.4 (1.7)	3.4 (1.6)	<0.001 *
	WORC	62.2 (15.0)	30.4 (14.5)	31.8 (14.2)	<0.001 *
Women	VAS	8.8 (1.7)	4.7 (1.7)	4.1 (1.2)	<0.001 *
	WORC	64.8 (13.8)	36.6 (17.2)	28.2 (16.6)	<0.001 *

SD: standard deviation; VAS: visual analogue scale; WORC: Western Ontario Rotator Cuff index. * *p* < 0.05.

**Table 3 healthcare-11-02786-t003:** Results from the correlation analysis.

Outcome	Overall		Men		Women	
	VAS	WORC	VAS	WORC	VAS	WORC
TSK						
Pearson’s Correlation Index	0.547	0.385	0.515	0.301	0.486	0.250
*p*-value	0.005 *	0.072	0.005*	0.081	0.012 *	0.110
PSEQ						
Pearson’s Correlation Index	−0.487	−0.212	−0.334	−0.116	−0.595	−0.080
*p*-value	0.001 *	0.217	0.068	0.285	0.001*	0.334
Educational level						
Spearman’s rho	−0.042	0.057	−0.030	0.061	−0.052	0.050
*p*-value	0.870	0.882	0.630	0.713	0.914	0.880
Household monthly income						
Spearman’s rho	−0.145	0.152	−0.181	0.160	−0.124	0.084
*p*-value	0.601	0.570	0.634	0.446	0.544	0.865
Job rank						
Spearman’s rho	0.121	0.110	0.125	0.115	0.098	0.085
*p*-value	0.670	0.632	0.670	0.601	0.840	0.878

PSEQ: Pain Self-Efficacy Questionnaire; TSK: Tampa Scale of Kinesiophobia; VAS: Visual Analogue Scale; WORC: Western Ontario Rotator Cuff index. * *p* < 0.05.

## Data Availability

The data that support the findings of this study are available upon reasonable request to the corresponding author.

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
