# Peer review of "Associations between Preoperative Patient Socioeconomic Status and Pain-Related Outcomes with Pain and Function in Patients Undergoing Rotator Cuff Repairs"

_healthcare, 2023, doi:10.3390/healthcare11202786_

Round 1

Reviewer 1 Report

Associations between preoperative patient socioeconomic status and pain-related outcomes with pain and function in patients undergoing rotator cuff repairs.

Thank you very much for allowing me to review this manuscript. I would like to provide some suggestions to the authors for its publication.

Methods:

- Please, specify how the sample size calculation was conducted and what statistical power was achieved.

- The objective has been mentioned at the end of the introduction and again in lines 105-108.

- The information provided in lines 126-128 should be placed in another section labeled "variables," or similar. I believe it should not be included in the "participants" section.

- The introduction addresses the concept of pain catastrophizing, but it is not evaluated using the scales mentioned in the "outcomes" section.

- Analyses should be conducted separately for women and men, as recommended by international guidelines.

- In Table 3, the models have not been adjusted for the variables mentioned in Table 1, is that correct? There are other variables that have not been considered, in addition to not indicating the values obtained for each category or providing model fit indicators.

- The discussion restates the results but does not delve into a thorough discussion.

Author Response

Thank you very much for allowing me to review this manuscript. I would like to provide some suggestions to the authors for its publication.

Authors would like to thank the Reviewer for the thorough review and the valuable suggestions. We have made the corresponding changes according to the given recommendations, and we would like to express our gratitude for being given the chance to improve our manuscript.

Methods:

Comment 1- Please, specify how the sample size calculation was conducted and what statistical power was achieved.

Response 1- Authors would like to thank the Reviewer for the comment, however, the information the Reviewer is asking for is provided in lines 175-181. We specified how the sample size was calculated and the statistical power achieved. Authors would like to apologize for the misunderstanding.

Comment 2- The objective has been mentioned at the end of the introduction and again in lines 105-108.

Response 2- We agree that redundancy should be avoided, therefore we have only mentioned the objective at the end of the Introduction and eliminated lines 105-108.

Comment 3- The information provided in lines 126-128 should be placed in another section labeled "variables," or similar. I believe it should not be included in the "participants" section.

Response 3- We agree with the Reviewer, and therefore we decided to move that information under the section “2.3. Outcomes”.

Comment 4- The introduction addresses the concept of pain catastrophizing, but it is not evaluated using the scales mentioned in the "outcomes" section.

Response 4- Authors would like to thank the Reviewer for the comment. Indeed, pain catastrophizing is introduced as a negative cognitive-affective response that might influence the experience of pain, however, our intention was to mention it only as an example. We agree with the Reviewer that this construct could have been assessed too, even though we eventually decided to include other constructs we believed were also relevant, such as kinesiophobia and pain self-efficacy. We have replaced the statement regarding pain catastrophizing, and we would like to apologize for the confusion.

Comment 5- Analyses should be conducted separately for women and men, as recommended by international guidelines.

Response 5- We have conducted the analyses separately for women and men, as suggested, and modified the reporting of results accordingly.

Comment 6- In Table 3, the models have not been adjusted for the variables mentioned in Table 1, is that correct? There are other variables that have not been considered, in addition to not indicating the values obtained for each category or providing model fit indicators.

Response 6- Our intention was to present variables such as smoking status and upper limb dominance with descriptive purposes, just to detail characteristics of the sample; it was not our intention to perform an analysis upon those variables, but we do understand the Reviewer’s concern. This is the same case for the variables presented in Table 1 regarding socioeconomic status; our intention was not to stratify educational level by categories and conduct the analysis within each category, our intention was to establish if there was a correlation between higher or lower educational levels and results for pain and function. This is why we chose to perform these analyses with Spearman’s rho instead of Pearson’s index, as the nature of the variables was not the same in either case.

Comment 7- The discussion restates the results but does not delve into a thorough discussion.

Response 7- Authors would like to thank the Reviewer for the suggestion, we understand that the Discussion is a key section within a manuscript and that it can always be improved. Indeed, we mention the results again but the intention is to discuss these results with previous studies to reinforce the key points presented in this manuscript. We discuss results which are similar to other studies and the implications that this might create, and we also mention results that differ from the known literature and discuss the reasons of why this might be. Additionally, we acknowledge several limitations our study has, including an additional limitation in the revised version, and we present the strengths too. Our intention is to highlight the influence of these preoperative factors in the prognosis of shoulder surgery. Nevertheless, we appreciate the Reviewer’s consideration and we thank the Reviewer for the thorough analysis.

Reviewer 2 Report

First of all, congratulations for the work done, then I will mention a number of changes and recommendations in order to obtain clearer and more accurate information.

- General comments:

- Comments on the introduction:

Lines 46-48. This statement needs a reference.

Western Ontario Rotator Cuff index must be abbreviated the first time it appears in the manuscript.

I do not see the point of talking about Western Ontario Rotator Cuff index to say that there are adapted scales if you apply it to a Spanish population. You should use a validated translated version of this index or at least a version in Spanish like the one used in Martinez-Cano et al. 2018. Or using another tested questionnaire like Rodriguez et al. 2020.

Martinez-Cano et al. 2018 (see link in PDF)

Rodriguez et al. 2020 (see link in PDF)

- Comments on material and methods

The Spanish version of the TSK is the TSK-11, then the score cannot vary from 17 to 68, the correct scoring varies from 11 to 44. You are talking about one TSK and using another different. Furthermore, you talk about a cutoff of 37 to determine the presence or absence of kinesiophobia, check this.

TSK, PSEQ, VAS, WORC… have been abbreviated in the introduction, you do not have to abbreviate them again. You must use the abbreviation directly.

- Comments on results:

You have done a one-way ANOVA to analyze the change over time, but in table 2 you talk about VAS in the pre-test but in the material and methods section, you did not indicate any measure of the VAS in the preoperative assessment.

- Comments on conclusions:

Again, if the abbreviations have been defined before, you must use them directly (lines 276-277).

Author Response

First of all, congratulations for the work done, then I will mention a number of changes and recommendations in order to obtain clearer and more accurate information.

Authors would like to thank the Reviewer for the insightful comments. We appreciate the suggestions and we hope to do our best to improve our manuscript according to the feedback.

- General comments:

- Comments on the introduction:

Comment 1- Lines 46-48. This statement needs a reference.

Response 1- We have included references for the opening statement.

Comment 2- Western Ontario Rotator Cuff index must be abbreviated the first time it appears in the manuscript.

Response 2- We would like to apologize, and we have amended the error by abbreviating the Western Ontario Rotator Cuff index the first time its mentioned in line 70 and only mentioning the abbreviation in line 168.

Comment 3- I do not see the point of talking about Western Ontario Rotator Cuff index to say that there are adapted scales if you apply it to a Spanish population. You should use a validated translated version of this index or at least a version in Spanish like the one used in Martinez-Cano et al. 2018. Or using another tested questionnaire like Rodriguez et al. 2020. Martinez-Cano et al. 2018 (see link in PDF) Rodriguez et al. 2020 (see link in PDF)

Response 3- Authors understand the Reviewer’s concern and would like to thank the Reviewer for the advice. When developing the design of the study, we were unable to retrieve a validated Spanish version in a Spanish population for the WORC, so we opted to use a Spanish translation of the original version, conducted by Kirkley et al. We wanted to avoid using validated translations in different cultural settings, such as the one presented in the study by Martinez-Cano et al. to ensure there were no cultural influences. We have added this as a limitation in our Discussion section and we have explicitly stated that we used a translation of the original version in our Materials and Method section. We would like to sincerely apologize for the inconveniences.

- Comments on material and methods

Comment 4- The Spanish version of the TSK is the TSK-11, then the score cannot vary from 17 to 68, the correct scoring varies from 11 to 44. You are talking about one TSK and using another different. Furthermore, you talk about a cutoff of 37 to determine the presence or absence of kinesiophobia, check this.

Response 4- Authors would like to sincerely apologize for the mistake. We initially intended to use the validated Spanish version of the TSK, developed by Gomez-Perez et al. However, we were unable to retrieve the actual scale from that publication. We opted to search online for the scale, and eventually came upon a TSK scale in Spanish, and now realize that it is a translation of the original English version rather than the validated Spanish version. We have modified our reporting according to this update, and added a statement in our Discussion section (lines 335-338) acknowledging our mistake as a limitation. Again, we would like to thank the Reviewer for pointing out this mistake and present our sincere apologies for it.

Comment 5- TSK, PSEQ, VAS, WORC… have been abbreviated in the introduction, you do not have to abbreviate them again. You must use the abbreviation directly.

Response 5- We have made the proper amendments and only abbreviate the terms the first time they appear in the text. We would like to thank the Reviewer for the comment.

- Comments on results:

Comment 6- You have done a one-way ANOVA to analyze the change over time, but in table 2 you talk about VAS in the pre-test but in the material and methods section, you did not indicate any measure of the VAS in the preoperative assessment.

Response 6-   We understand that our reporting might have led to confusion, so we have modified the names of the subsections in Materials and Method to be clear. Along with what we first named “preoperative assessments” we also assessed pain intensity and function, so we have re-named the subsections as “Pre and post-surgical assessments” to include pain intensity and function, and “Outcomes for the correlation analysis” to include fear of movement, pain self-efficacy, and the participant’s socioeconomic status.

- Comments on conclusions:

Comment 7- Again, if the abbreviations have been defined before, you must use them directly (lines 276-277).

Response 7- We have presented only the abbreviations of the terms that have been abbreviated earlier in the manuscript. Again, we would also like to apologize for this matter.

Round 2

Reviewer 1 Report

Thank you very much for allowing me to review the manuscript once again. The authors have done a commendable job and have addressed all my doubts and concerns regarding the manuscript.

Reviewer 2 Report

All my suggestions have been amended.